An efficient gradient-based algorithm with descent direction for unconstrained optimization with applications to image restoration and robotic motion control

http://orcid.org/0000-0001-5246-6636 Ibrahim Sulaiman Mohammed 1 2
Awwal Aliyu M. 3
Malik Maulana 4
Khalid Ruzelan 1
Benjamin Aida Mauziah 1 mauziah@uum.edu.my
Mohd Nawawi Mohd Kamal 1
http://orcid.org/0000-0001-5557-2231 Madi Elissa Nadia 5
1 School of Quantitative Sciences, Universiti Utara Malaysia , Sintok, Kedah , Malaysia
2 Faculty of Arts and Education, Sohar University , Sohar , Oman
3 Department of Mathematics, Gombe State University , Gombe , Nigeria
4 Department of Mathematics, Faculty of Mathematics and Natural Sciences, Universitas Indonesia , Depok , Indonesia
5 Faculty of Informatics and Computing, Universiti Sultan Zainal Abidin , Besut, Terengganu , Nigeria
Coelho Paulo Jorge
Electronic publication date: 2025 May 23
Publication date: 2025
Volume: 11
Electronic Location ID: e2783
Received 2024 Aug 30; Accepted 2025 Mar 4
Copyright: © 2025 Ibrahim et al.
Copyright year: 2025
Copyright holder: Ibrahim et al.
License: This is an open access article distributed under the terms of the Creative Commons Attribution License, which permits unrestricted use, distribution, reproduction and adaptation in any medium and for any purpose provided that it is properly attributed. For attribution, the original author(s), title, publication source (PeerJ Computer Science) and either DOI or URL of the article must be cited.
License URL: https://creativecommons.org/licenses/by/4.0/

Keywords: Gradient based method, Image restoration, Robotic motion control, Unconstrained optimization, Convergence analysis

Funding: Ministry of Higher Education (MoHE) of Malaysia FRGS/1/2022/STG06/UUM/02/6 This research was supported by Ministry of Higher Education (MoHE) of Malaysia through Fundamental Research Grant Scheme (FRGS/1/2022/STG06/UUM/02/6). The funders had no role in study design, data collection and analysis, decision to publish, or preparation of the manuscript.

==============================
This study presents a novel gradient-based algorithm designed to enhance the performance of optimization models, particularly in computer science applications such as image restoration and robotic motion control. The proposed algorithm introduces a modified conjugate gradient (CG) method, ensuring the CG coefficient, β κ, remains integral to the search direction, thereby maintaining the descent property under appropriate line search conditions. Leveraging the strong Wolfe conditions and assuming Lipschitz continuity, we establish the global convergence of the algorithm. Computational experiments demonstrate the algorithm’s superior performance across a range of test problems, including its ability to restore corrupted images with high precision and effectively manage motion control in a 3DOF robotic arm model. These results underscore the algorithm’s potential in addressing key challenges in image processing and robotics.

Introduction

Unconstrained optimization plays a critical role in various computer science and engineering applications, including image processing (Yuan, Lu & Wang, 2020; Awwal et al., 2023), signal recovery (Wu et al., 2023), machine learning (Kamilu et al., 2023; Kim et al., 2023), and robotic control systems (Awwal et al., 2021; Yahaya et al., 2022). These applications often involve the optimization of complex objective functions, where robust and efficient numerical formulations are essential for achieving high performance. Among the many optimization methods, the conjugate gradient (CG) algorithm has garnered significant attention due to its balance between computational efficiency and convergence properties for large-scale minimization problem of the form:

(1) minf(x),x∈Rn,

where f:Rn→R is a smooth function, and its gradient g(x)=∇f(x) is available (Hager & Zhang, 2006; Ivanov et al., 2023; Sabi’u et al., 2024; Awwal & Botmart, 2023; Salihu et al., 2023b; Sulaiman et al., 2024). One of the major problems that researchers need to tackle when minimizing Eq. (1) is identifying the best iterative procedure that will produce optimal values of x (Powell, 1977). In fact, a typical approach is to maintain a list of active points, which may at first be an initial guess, and to amend this list as the computation progresses. The major components of the computations include minimization of Eq. (1) and updating the iterative points as the calculations proceed.

The CG method is an iterative algorithm that begins with an initial guess x0∈Rn and proceeds to generate a succession of iterates using:

(2) xk+1:=xk+αkdk.k≥0,

with αk>0 defining the step size, which is computed along a direction of search dk∈Rn (Ibrahim & Mamat, 2020). The step size αk is often computed using either exact or inexact line search techniques (Salihu et al., 2024). However, most of the recent studies consider the inexact procedure because it is less competitive and produces approximate values of the step size (Hager & Zhang, 2006). The line search condition considered in this study is the weak Wolfe Powell (WWP), which computes αk such that:

(3) f(xk+αkdk)≤f(xk)+δαkgkTdk,

(4) |g(xk+αkdk)Tdk|≤−σgkTdk,

with gk=g(xk) and 0<δ<σ<1 (see Sun & Yuan, 2006; Wolfe, 1969, 1971).

One of the significant components of the CG iterative Formula (2) is dk, which is computed as:

(5) d0=−g0,dk=−gk+βkdk−1,k≥1,

where the scalar βk is known as the CG parameter, which characterizes the different CG formulas (Malik et al., 2023; Salihu et al., 2023a). Some classical nonlinear CG algorithms are presented by Hestenes & Stiefel (1951) (HS), Polak & Ribiere (1969), Polyak (1967) (PRP), and Liu & Storey (1991) (LS) with the following βk updating formula:

(6) βkHS=gkT(gk−gk−1)dk−1T(gk−gk−1),βkPRP=gkT(gk−gk−1)∥gk−1∥2,βkLS=−gkT(gk−gk−1)dk−1Tgk−1,

with ℓ2 norm given as ∥⋅∥. These classical CG formulas are computationally efficient but can sometimes fail to achieve global convergence for general functions (Hager & Zhang, 2006). For instance, Powell identified issues with the PRP formula cycling without reaching an optimum, even with line search techniques (Yao, Zengxin & Hai, 2006). Another class of classical CG algorithms is presented by Fletcher & Powell (1963) (FR), Dai & Yuan (1999) (DY), and Fletcher (1987) (CD) with the following formulas:

(7) βkFR=||gk||2||gk−1||2,βkDY=||gk||2dk−1T(gk−gk−1),βkCD=−||gk||2dk−1Tgk−1.

Unlike the first category of the classical CG method presented in Eq. (6), the class of FR, DY, and CD methods is characterized by strong convergent properties. However, their performance is affected by jamming phenomena (Hager & Zhang, 2006; Deepho et al., 2022).

To address these limitations, researchers have developed modifications to improve the convergence and robustness of the CG formulas. One notable modification of the PRP method restricts βk to non-negative values, resulting in the PRP+ variant:

(8) βkPRP+=max{βkPRP,0}.

This formula improved the computational efficiency as well as the convergence results of the PRP formula. The convergence properties of Eq. (8) were further explored under suitable conditions (Gilbert & Nocedal, 1992; Powell, 1986). Additional modifications to enhance the robustness of the CG methods include the Enhanced PRP (EPRP) formula by Babaie-Kafaki & Ghanbari (2014), which introduces a parameter ω≥0:

(9) βkEPRP=βkPRP−ωgkTdk−1||gk−1||2,ω≥0.

When ω=0, this formula reduces to the classical PRP method defined in Eq. (6). Babaie-Kafaki & Ghanbari (2017) later extended this modification by using βkPRP+ instead of βkPRP, creating the EPRP+ variant:

(10) βkEPRP+=βkPRP+−ωgkTdk−1||gk−1||2,ω≥0.

The new modification possesses a relatively good numerical performance and the convergence was discussed under mild assumption. For more reference on modifications of the classical CG methods, (see Babaie-Kafaki, Mirhoseini & Aminifard, 2023; Yao, Zengxin & Hai, 2006; Hager & Zhang, 2006; Zengxin, Shengwei & Liying, 2006; Hai & Suihua, 2014; Awwal et al., 2023; Li & Du, 2019; Yu, Kai & Xueling, 2023; Ibrahim & Salihu, 2025; Shao et al., 2023; Jian et al., 2022; Wu et al., 2023; Malik et al., 2021)

Other modifications include the Rivaie–Mustafa–Ismail–Leong method (RMIL) variant proposed by Rivaie et al. (2012b), which revises the Hestenes–Stiefel method (HS) denominator to enhance convergence:

(11) βkRMIL=gkT(gk−gk−1)dk−1T(dk−1−gk).

In Eq. (11), the authors replaced the term (gk−gk−1) in the denominator of the HS formula with (dk−1−gk) and established the convergence under some mild assumptions. It is obvious that the numerator of Eq. (11) is the same as that of the PRP method, thus, the computation results of this method were evaluated using the classical HS and PRP Eq. (6). Rivaie et al. (2012a) extended the approach presented in Eq. (11) to define a new formula as follows:

(12) βkRMIL∗=gkT(gk−gk−1)||dk−1||2,

and further simplified Eq. (12) to present another modification known as RMIL+ (Rivaie, Mamat & Abdelrhaman, 2015) with the formula given as:

(13) βkRMIL+=gkT(gk−gk−1−dk−1)||dk−1||2,

The convergence analysis of these methods was discussed based on the following simplification:

(14) 0≤βk≤||gk||2||dk−1||2.

where βk follows from Eqs. (12) and (13). The inequality Eq. (14) is very significant in discussing the convergence of the above RMIL formulas. However, a note from Dai (2016) raised some concern regarding Eq. (14) which invalidate the convergence results of the RMIL formula and further corrected the formula by imposing the restrictions 0≤gkTgk−1≤||gk||2 to the RMIL ∗ CG coefficient Eq. (12). The introduction of this inequality by Dai (2016) has led to several variants of the RMIL-type methods (see; (Yousif, 2020; Awwal et al., 2021; Sulaiman et al., 2022)). These modifications are constructed based on the restriction imposed on the RMIL formula and their convergence hugely depends on the above condition. However, it is obvious that the modified RMIL might become redundant if the inner product gkTgk−1 is negative or bigger than or equal to ||gk||2, and the search directions associated with them will reduce to the classical steepest descent. A notable drawback of the steepest descent method is its tendency to converge slowly, especially in ill-conditioned problems, as it often oscillates in narrow valleys of the objective function landscape, making it inefficient for large-scale optimization. More so, many of the CG methods available in literature face challenges, particularly in maintaining a descent direction throughout iterations, which is crucial for ensuring convergence in non-linear optimization problems.

This study addresses the identified limitations by designing a new conjugate gradient formula in such a way that the restriction imposed by Dai (2016) is avoided and guarantees the sufficient descent condition. The proposed algorithm incorporates a refined descent direction condition, ensuring robust performance across various test cases and achieving better global convergence properties under the strong Wolfe Powell (SWP) line search conditions. The modified search direction is especially advantageous in situations where classical methods tend to exhibit instability or cycling behavior, as it effectively mitigates these issues while preserving computational efficiency.

The remaining sections of this study outline the formulation of the proposed algorithm and provide a detailed convergence analysis. We also present extensive computational results demonstrating the efficacy of the proposed algorithm in restoring degraded images, handling dynamic motion control in robotic systems, and solving unconstrained optimization problems across diverse domains. These findings highlight the algorithm’s potential and versatility for broader application in complex optimization situations, suggesting promising future research directions in both applied and theoretical optimization fields.

Materials and Methods

The study by Rivaie et al. (2012a) claimed that Eq. (14) holds for all k≥1. However, Dai (2016) countered that assertion by showing that gkTgk−1≥0 is not guaranteed for all k. This implies that condition Eq. (14) cannot generally hold for the βkRMIL defined in Eq. (12). Dai (2016) presented a modification as follows:

(15) βkRMIL+∗={gkT(gk−gk−1)||dk−1||2,if0≤gkTgk−1≤||gk||2,0,otherwise,

and further discussed global convergence using suitable assumptions. It is worth noting that the convergence of different variants of βkRMIL+∗ hugely relies on 0≤gkTgk−1≤||gk||2.

Remark 0.1 From Eq. (15), it is obvious that the coefficient βkRMIL+∗ will likely become superfluous and its corresponding dk reduces to the well-known steepest descent direction, that is, dk=−gk, if the inner product gkTgk−1 is negative or greater than or equal to ||gk||2. These are some of the drawbacks associated with the βkRMIL+∗.

From the above discussion, it is obvious to see that the βkRMIL+∗ formula has been restricted by the condition 0≤gkTgk−1≤||gk||2 which may not hold for general functions. To address this issue, we present the following variant of βkRMIL+∗:

(16) βkSRMIL=gkT(gk−gk−1)||dk−1||2−θgkTdk−1||gk−gk−1||||dk−1||4,

and the new dk is given as:

(17) d0=−g0,dk=−gk+1γk(βkSRMILdk−1−βkSRMILgkdk−1Tgk||gk||2)

where θ>0 and γk=βkSRMIL||dk−1||μ||gk||,0<μ<1.

The following algorithm describes the execution procedure of the proposed method.

Convergence analysis

The assumption that follows would be important in studying the convergence analysis of the new CG algorithm.

Assumption 0.2 (i) The underlying function, f(x), is bounded below on the level set Γ={x∈Rn|f(x0)≥f(x)}.

(ii) Denoting Γ^ as some neighbourhood of Γ which is open and convex, then f is smooth and g(x)=∇f(x) satisfies the Lipchitz continuity on Γ^⊇Γ, that is,

(19) ||g(x)−g(y)||≤L||x−y||,∀x,y∈Γ^,L>0.

Remark 0.3 Based on the Assumption 0.2, it is not difficult to see that the following conclusions hold

(20) ||g(x)||≤τ,∀x∈Γ,τ>0.

(21) ||x−y||≤b,∀x,y∈Γ,b>0.

Since f(x) is a decreasing function and Assumption 0.2 shows that {xk} obtained using the proposed scheme is contained in a bounded region, then it follows that {xk} is convergent.

In what follows, we establish that the sequence {dk} produced by Algorithm 1 is sufficiently descending.

Algorithm 1 SRMIL algorithm.

  Input: Initialization: x0∈Rn,0<μ<1,θ>0, Termination tolerance ε>0.	
  Step 1: Obtain gk, if ||gk||≤0, then	
     terminate the iteration process.	
  end	
  Step 2: k=0 or βkSRMIL≤0, dk:=−gk;	
(18) dk=−gk+1γk(βkSRMILdk−1−βkSRMILgkdk−1Tgk||gk||2)

	
        with the parameter βkSRMIL being determined using Eq. (16) and γk=βkScaledRMILmethod(SRMIL)||dk−1||μ||gk||.	
  Step 3:Determine αk such that Eqs. (3) and (4) are satisfied.	
  Step 4: Calculate new point using Eq. (2).	
  Step 5: Return to Step 1 with k:=k+1.	

Lemma 0.4 The sequence the {dk} from Algorithm 1 is sufficiently descent, that is:

(22) gkTdk≤−c||gk||2,

holds for all k.

Proof

gkTdk=−||gk||2+1γk(βkSRMILgkTdk−1−βkSRMIL(gkTdk−1)2||gk||2)≤−||gk||2+μ||gk||βkSRMIL||dk−1||βkSRMIL||gk||||dk−1||=−||gk||2+μ||gk||2=−(1−μ)||gk||2=−μ¯||gk||2

Note: since 0<μ<1, then (1−μ) is positive and hence, μ¯=1−μ is also positive.

The lemma that follows is crucial to the convergence of the new formula and can be found in the reference (Zoutendijk, 1970).

Lemma 0.5 Suppose Assumption 0.2 holds and dk is sufficiently descending with αk being determined by Eqs. (3) and (4). Then

(23) ∑k=0∞(gkTdk)2||dk||2<+∞.

Remark 0.6 It has been shown in Lemma 0.4 that dk obtained by Eq. (18) is sufficiently descending. Furthermore, αk is computed by Eqs. (3) and (4). Hence the condition Eq. (23) holds for the proposed Algorithm 1.

Now, we prove the convergence result of the new formula.

Theorem 0.7 If Assumption 0.2 is true and {xk} is produced by Algorithm 1, then

(24) limk→∞⁡inf||gk||=0.

Proof If Eq. (24) is not true, there will exist some constant c>0 for which

(25) ||gk||≥c,k≥0.

We first show that there is a constant τ^>0 satisfying:

(26) ||dk||≤τ^.

For k≥1 and βkSRMIL>0, then dk defined in Eq. (18) becomes

dk=−gk+1γk(βkSRMILdk−1−βkSRMILgkdk−1Tgk||gk||2),

and therefore,

||dk||≤||gk||+||1γk(βkSRMILdk−1−βkSRMILgkdk−1Tgk||gk||2)||≤||gk||+μ||gk|||βkSRMIL|||dk−1|||βkSRMIL|||(dk−1−gkdk−1Tgk||gk||2)||≤||gk||+μ||gk||||dk−1||(||dk−1||+||gk||2||dk−1||||gk||2)≤(1+2μ)||gk||≤(1+2μ)τ.

Hence, since τ<(1+2μ)τ, then setting τ^:=(1+2μ)τ yields Eq. (26).

Furthermore, by using Eqs. (22), (25) and (26), we get

∑k=0∞(gkTdk)2||dk||2≥∑k=0∞||gk||4||dk||2≥∑k=0∞c4τ^2=+∞.

This is a contradiction with Eq. (23). Hence, Eq. (24) holds.

Results

Unconstrained optimization problems

This section evaluates the efficiency of the new Scaled RMIL method (SRMIL) formula on some test functions taken from Andrei (2008) and Bongartz et al. (1995). The study considered a minimum of two numerical experiments for each problem with the variables varying from 2 to 1,000,000. The results of the proposed algorithm were compared with formulas with similar characteristics from RMIL method (Rivaie et al., 2012a), RMIL+ method (Dai, 2016), spectral Jin–Yuan–Jiang–Liu–Liu method (JYJLL) (Jian et al., 2020), New Three-Term Conjugate Gradient (NTTCG) method (Guo & Wan, 2023), and Conjugate Gradient (CG) DESCENT method (Hager & Zhang, 2005). Each method is coded in MATLAB R2019a and compiled on a PC with the specifications of Intel Core i7 CPU with 32.0 GB memory. The method is implemented under the Weak Wolfe Powell (WWP) conditions Eqs. (3) and (4) with values σ=0.1 and δ=0.01. To stop execution, we use the same criteria ||gk||∞≤10−6, or iteration number is ≥10,000, with ||gk||∞ denoting the maximum absolute of the gradient at kth iteration. If an algorithm fails to satisfy the stopping criteria, it will be considered a failure, and the point of failure will be denoted by (***). The experiments result based on CPU time (Tcpu), Number of function evaluation (NF), and iterations (Itr) is presented in Tables 1, 2, 3, 4. For clarity, we bolded the best results, i.e., the lowest number of iterations, CPU time, and function evaluations, respectively, to easily differentiate the performance of the algorithms.

Table 1 Performance Comparison Based on CPU time (Tcpu), Number of function evaluation (NF), and number of iterations (Itr).

		RMIL	RMIL+	SRMIL	JYJLL	bib14	CG_DESCENT	
S/N	Functions/Dimension	Tcpu	NF	Itr	Tcpu	NF	Itr	Tcpu	NF	Itr	Tcpu	NF	Itr	Tcpu	NF	Itr	Tcpu	NF	Itr	
P1	COSINE 6,000	1.08E−01	198	40	1.31E−01	181	35	7.14E−02	108	40	0.136	103	33	9.74	194	81	0.13	80	20	
P2	COSINE 100,000	1.78E+00	307	77	1.23E+00	218	46	6.57E−01	123	41	2.885	374	184	***	***	***	1.081	148	74	
P3	COSINE 800,000	9.85E+01	1,887	449	8.99E+01	1,703	358	4.85E+00	119	30	10.3	170	55	***	***	***	5.932	105	35	
P4	DIXMAANA 2,000	3.51E−01	178	36	3.92E−01	192	36	1.73E−01	82	18	0.229	83	20	2.36	85	19	0.175	83	27	
P5	DIXMAANA 30,000	7.04E+00	242	45	7.42E+00	233	39	2.16E+00	90	23	2.93	89	20	***	***	***	2.425	90	28	
P6	DIXMAANB 8,000	2.08E+00	223	41	1.95E+00	225	37	7.97E−01	86	20	1.005	93	24	41.1	96	22	0.594	80	25	
P7	DIXMAANB 16,000	3.04E+00	188	32	3.28E+00	202	37	1.11E+00	84	16	1.66	93	24	453	91	20	1.243	87	25	
P8	DIXMAANC 900	1.88E−01	210	40	2.03E−01	212	40	3.10E−01	92	23	0.252	89	25	0.59	80	18	0.066	74	23	
P9	DIXMAANC 9,000	2.60E+00	256	49	2.73E+00	262	49	7.35E−01	92	18	0.763	73	10	36.1	88	16	0.7	88	31	
P10	DIXMAAND 4,000	8.84E−01	190	39	1.11E+00	234	46	4.49E−01	93	26	0.477	90	21	8.7	88	19	0.329	87	26	
P11	DIXMAAND 30,000	9.23E+00	321	58	5.22E+00	178	28	2.33E+00	93	22	2.791	87	21	***	***	***	2.841	113	45	
P12	DIXMAANE 800	2.37E+00	2,861	783	3.20E+00	3,939	790	1.33E+00	1,928	1,226	***	***	***	8.19	706	358	0.739	837	719	
P13	DIXMAANE 16,000	***	***	***	***	***	***	***	***	***	***	***	***	***	***	***	***	***	***	
P14	DIXMAANF 5,000	***	***	***	***	***	***	3.96E+01	11,275	7,164	2.357	2,679	1,535	475	1,010	516	7.104	1,643	1,442	
P15	DIXMAANF 20,000	***	***	***	***	***	***	***	***	***	3.375	3,232	1,887	***	***	***	***	***	***	
P16	DIXMAANG 4,000	2.52E+01	5,269	1,507	2.32E+01	5,175	1,083	3.12E+01	10,387	6,858	***	***	***	***	***	***	6.063	1,469	1,333	
P17	DIXMAANG 30,000	***	***	***	***	***	***	***	***	***	***	***	***	***	***	***	***	***	***	
P18	DIXMAANH 2,000	***	***	***	1.77E+01	8,320	1,835	1.68E+01	8,808	6,862	***	***	***	***	***	***	1.962	1,019	928	
P19	DIXMAANH 50,000	***	***	***	***	***	***	***	***	***	***	***	***	***	***	***	***	***	***	
P20	DIXMAANI 120	***	***	***	***	***	***	***	***	***	***	***	***	***	***	***	***	***	***	
P21	DIXMAANI 12	1.75E−01	1,145	331	6.77E−02	1,358	287	4.13E−01	1,549	1,019	0.67	2,479	1,465	0.06	467	244	0.029	591	490	
P22	DIXMAANJ 1,000	***	***	***	***	***	***	***	***	***	***	***	***	***	***	***	0.029	591	490	
P23	DIXMAANJ 5,000	***	***	***	***	***	***	1.58E+01	2,327	1,715	15.52	2,239	1,241	***	***	***	***	***	***	
P24	DIXMAANK 4,000	2.92E+01	6,315	1,857	***	***	***	5.39E+00	1,336	999	7.614	1,490	829	***	***	***	2.45	663	575	
P25	DIXMAANK 40	1.60E−01	2,061	579	2.81E−01	3,570	793	2.98E−01	3,591	2,553	***	***	***	0.29	1,255	647	0.119	1,214	1,060	
P26	DIXMAANL 800	***	***	***	3.79E+01	1,8606	4,447	***	***	***	***	***	***	15.3	1,118	564	0.983	1,214	1,060	
P27	DIXMAANL 8,000	1.20E+02	8,138	2,635	8.03E+01	6,284	1,595	3.65E+00	385	218	14.54	1,539	853	***	***	***	10.54	1,579	1,382	
P28	DIXON3DQ 150	***	***	***	1.04E+00	39,572	9,689	***	***	***	***	***	***	0.16	2,162	1,374	***	***	***	
P29	DIXON3DQ 15	3.79E−02	895	270	4.47E−02	1,322	349	3.54E−02	741	471	0.204	740	422	0.03	222	144	0.006	284	226	
P30	DQDRTIC 9,000	2.07E−01	551	133	1.79E−01	713	159	1.97E−01	684	397	0.285	822	446	22.5	213	72	0.069	229	134	
P31	DQDRTIC 90,000	4.99E−01	549	135	7.41E−01	808	172	9.93E−01	862	544	2.059	776	414	***	***	***	0.237	216	124	
P32	QUARTICM 5,000	3.92E−01	220	50	3.78E−01	249	59	2.42E−01	152	46	0.247	143	42	***	***	***	0.245	156	62	
P33	QUARTICM 150,000	3.05E+01	399	81	1.65E+01	319	68	4.04E+01	456	126	11.68	250	84	***	***	***	10.27	254	121	
Note:

*** point of failure. Best results are in bold.

Table 2 Performance Comparison Based on CPU time (Tcpu), Number of function evaluation (NF), and number of iterations (Itr).

		RMIL	RMIL+	SRMIL	JYJLL	bib14	CG_DESCENT	
S/N	Functions/Dimension	Tcpu	NF	Itr	Tcpu	NF	Itr	Tcpu	NF	Itr	Tcpu	NF	Itr	Tcpu	NF	Itr	Tcpu	NF	Itr	
P34	EDENSCH 7,000	1.09E+00	442	80	9.85E−01	447	75	3.10E−01	346	70	0.348	165	43	17	578	80	0.207	100	40	
P35	EDENSCH 40,000	1.59E+01	937	121	2.18E+01	2,203	258	1.36E+00	377	70	2.302	192	44	***	***	***	11.26	1,050	132	
P36	EDENSCH 500,000	4.13E+02	1,082	137	7.51E+02	6,132	655	1.68E+01	235	53	89.65	603	85	***	***	***	32.35	242	61	
P37	EG2 100	6.87E−01	14,686	2,619	7.03E−01	25,756	4,804	***	***	***	***	***	***	0.03	697	191	0.247	8,864	1,936	
P38	EG2 35	1.07E−01	3,885	1,037	7.69E−02	3,664	773	1.86E−01	6,480	4,780	***	***	***	0.16	445	137	0.024	875	614	
P39	FLETCHCR 1,000	2.83E−01	1,362	171	1.10E−01	1,678	200	1.59E−02	237	134	0.009	207	116	0.34	469	71	0.101	3,671	399	
P40	FLETCHCR 50,000	9.66E+00	10,270	1,024	9.54E+00	10,490	1,111	3.23E-01	245	132	0.287	235	109	***	***	***	3.674	4,831	527	
P41	FLETCHCR 200,000	3.86E+01	7,771	768	1.07E+01	4,010	446	1.63E+00	458	189	13.53	3,144	321	***	***	***	13.68	4,941	592	
P42	Freudenstein & Roth 460	7.83E-01	19,998	2,336	7.51E-01	23,036	2,597	1.74E+00	62,438	8,258	***	***	***	1.75	14,067	1,371	***	***	***	
P43	Freudenstein & Roth 10	9.16E−02	3,320	797	1.10E−01	5,381	964	8.52E−02	3,458	1,730	***	***	***	0.08	676	156	0.011	585	237	
P44	Generalized Rosenbrock 10,000	***	***	***	***	***	***	***	***	***	***	***	***	***	***	***	***	***	***	
P45	Generalized Rosenbrock 100	2.37E−01	7,453	2,507	1.70E−01	7,985	2,452	3.62E−01	13,459	9,499	***	***	***	0.06	1,073	681	0.043	1,843	1,741	
P46	HIMMELBG 70,000	9.14E−02	15	2	1.11E−01	15	2	4.58E−02	15	2	0.043	15	2	***	***	***	0.1	26	6	
P47	HIMMELBG 240,000	1.63E−01	24	3	1.32E−01	24	3	1.01E−01	13	2	0.68	13	2	***	***	***	0.258	30	6	
P48	LIARWHD 15	6.42E−02	174	40	4.84E−02	169	34	9.24E−03	145	61	0.008	219	117	0.01	208	58	0.022	155	92	
P49	LIARWHD 1,000	4.62E−02	824	179	3.07E−02	870	167	2.96E−01	7,766	5,719	0.047	1,428	836	0.32	360	72	0.016	495	269	
P50	Extended Penalty 1,000	2.82E+00	1,531	205	1.00E+00	624	89	2.92E−01	211	28	0.331	212	32	0.25	231	33	0.081	94	26	
P51	Extended Penalty 8,000	***	***	***	***	***	***	7.94E+00	93	16	10.99	93	16	11.6	142	23	3.801	112	27	
P52	QUARTC 4,000	3.95E−01	258	65	6.48E−01	269	65	1.92E−01	175	49	0.19	156	50	15.3	468	241	0.175	145	54	
P53	QUARTC 80,000	9.29E+00	457	88	1.01E+01	414	81	5.17E+00	251	85	6.926	263	89	***	***	***	5.098	242	117	
P54	QUARTC 500,000	8.22E+01	638	133	1.84E+02	524	121	4.81E+01	368	131	51.31	323	109	***	***	***	40.89	311	167	
P55	TRIDIA 300	2.95E−01	7,563	2,513	1.24E+00	10,661	2,506	1.71E−01	5,899	4,240	***	***	***	0.11	908	596	0.057	1,340	1,193	
P56	TRIDIA 50	2.78E−02	1,214	408	7.17E−02	2,379	566	5.69E−02	1,892	1,239	0.282	2,028	1,187	0.02	363	231	0.012	530	463	
P57	Extended Woods 150,000	3.73E+00	1,912	518	5.41E+00	2,348	576	9.15E+00	3,146	2,110	***	***	***	***	***	***	1.369	600	439	
P58	Extended Woods 200,000	4.86E+00	2,103	588	9.10E+00	2,052	533	8.89E+00	2,465	1,651	***	***	***	***	***	***	1.388	480	332	
P59	BDEXP 5,000	5.93E−02	11	2	2.13E−01	11	2	5.55E−02	11	2	0.007	11	2	0.1	18	2	0.043	22	6	
P60	BDEXP 50,000	5.37E−02	16	2	7.00E−02	16	2	5.01E−02	16	2	0.059	16	2	28.1	11	2	0.17	37	9	
P61	BDEXP 500,000	4.19E−01	12	2	5.80E−01	12	2	4.30E−01	12	2	2.157	12	2	***	***	***	1.112	27	7	
P62	DENSCHNF 90,000	4.05E−01	225	41	5.89E−01	289	53	1.43E−01	108	27	0.154	102	24	***	***	***	0.162	105	31	
P63	DENSCHNF 280,000	1.27E+00	330	54	1.29E+00	302	58	4.78E−01	105	26	0.58	106	23	***	***	***	0.442	108	32	
P64	DENSCHNF 600,000	2.23E+00	281	49	4.16E+00	285	48	1.04E+00	115	25	1.508	111	25	***	***	***	0.951	110	31	
P65	DENSCHNB 6,000	1.73E−01	223	38	1.94E−01	191	32	1.77E−02	82	20	0.009	86	21	2.06	79	18	0.032	66	19	
P66	DENSCHNB 24,000	7.57E−02	201	35	1.03E−01	229	35	3.68E−02	84	20	0.042	90	21	32.3	80	18	0.021	79	21	
Note:

*** point of failure. Best results are in bold.

Table 3 Performance Comparison Based on CPU time (Tcpu), Number of function evaluation (NF), and number of iterations (Itr).

		RMIL	RMIL+	SRMIL	JYJLL	bib14	CG_DESCENT	
S/N	Functions/Dimension	Tcpu	NF	Itr	Tcpu	NF	Itr	Tcpu	NF	Itr	Tcpu	NF	Itr	Tcpu	NF	Itr	Tcpu	NF	Itr	
P67	Extended DENSCHNB 300,000	7.62E−01	240	43	8.40E−01	247	37	3.38E−01	97	21	1.478	91	19	***	***	***	0.287	88	24	
P68	Generalized Quartic 9,000	2.26E−01	248	44	1.50E−01	136	27	2.51E−02	70	14	0.031	83	19	5.16	84	19	0.042	84	28	
P69	Generalized Quartic 90,000	2.43E−01	223	40	2.60E−01	196	40	1.05E−01	89	18	0.138	82	16	***	***	***	0.088	83	23	
P70	Generalized Quartic 500,000	1.78E+00	254	48	2.27E+00	270	46	1.64E+00	196	88	1.141	101	25	***	***	***	0.618	89	23	
P71	BIGGSB1 110	3.14E−01	6,643	2,247	2.03E−01	5,133	1,240	1.52E−01	5,507	3,964	0.09	2,890	1,684	0.04	719	462	0.029	858	767	
P72	BIGGSB1 200	3.99E−01	18,124	6,62	5.50E−01	22,895	5,399	***	***	***	***	***	***	0.15	1,547	986	0.03	1,425	1,365	
P73	SINE 100,000	3.43E+01	6,087	1,426	8.95E+00	1,122	245	8.99E−01	149	48	1.448	192	61	***	***	***	***	***	***	
P74	SINE 50,000	2.15E+01	7,331	1,714	1.44E+01	3,724	857	3.32E-01	105	33	0.969	168	59	***	***	***	***	***	***	
P75	FLETCBV 15	1.16E−01	235	68	1.80E−01	263	89	1.14E−02	217	121	0.007	195	91	***	***	***	0.022	69	36	
P76	FLETCBV 55	1.03E−01	2,374	1,150	1.99E−01	4,642	1,485	1.04E−01	2,616	2,012	1.066	2,148	1,425	***	***	***	0.018	403	299	
P77	NONSCOMP 5,000	1.70E−01	348	79	2.73E−01	294	76	1.95E−02	130	57	0.016	118	45	5.41	145	63	0.047	140	66	
P78	NONSCOMP 80,000	6.88E−01	602	114	6.21E−01	538	111	1.90E−01	164	75	0.501	203	87	***	***	***	0.149	137	66	
P79	POWER 150	***	***	***	8.62E−01	35,232	8,887	***	***	***	0.011	423	239	0.14	2,437	1,554	***	***	***	
P80	POWER 90	3.54E−01	15,279	5,004	3.00E−01	13,552	3,444	***	***	***	0.085	3,102	1,817	0.06	1,489	985	***	***	***	
P81	RAYDAN1 500	5.80E−02	1,220	346	8.73E−02	1,729	397	3.68E−02	869	559	0.044	1,266	713	0.27	452	262	0.035	497	415	
P82	RAYDAN1 5,000	***	***	***	***	***	***	1.47E+00	13,521	5,482	***	***	***	92.1	1,716	966	***	***	***	
P83	RAYDAN2 2,000	2.48E−02	136	20	4.24E−02	136	20	9.29E−03	71	14	0.005	71	14	0.22	75	16	0.201	1,566	168	
P84	RAYDAN2 20,000	7.99E−02	129	23	1.27E−01	175	28	7.15E−02	97	15	0.11	97	15	30.6	90	25	0.595	516	68	
P85	RAYDAN2 500,000	6.15E+00	669	78	7.86E+00	628	72	1.43E+00	139	23	3.898	251	37	***	***	***	20.78	777	94	
P86	DIAGONAL1 800	2.23E+00	52,409	5,145	1.43E+00	33,782	3,379	5.23E−01	12,030	1,520	0.216	4,004	904	3.51	8,392	966	***	***	***	
P87	DIAGONAL1 2,000	4.47E+00	60,332	6,001	3.90E+00	49,500	5,145	2.63E+00	3,266	4,586	***	***	***	***	***	***	***	***	***	
P88	DIAGONAL2 100	2.54E−02	288	86	4.55E−02	379	104	1.55E−02	272	155	0.01	265	133	0.01	239	115	0.017	165	117	
P89	DIAGONAL2 1,000	6.66E−02	1,186	362	1.08E−01	1,351	329	7.44E−02	1,264	827	0.231	1,696	962	1.99	970	433	0.062	603	516	
P90	DIAGONAL3 500	3.98E−01	7,112	822	4.81E−01	8,946	1,052	1.95E−01	3,900	836	0.229	3,825	1,068	0.81	3,984	527	***	***	***	
P91	DIAGONAL3 2,000	***	***	***	***	***	***	3.20E+00	24,679	3,928	***	***	***	***	***	***	***	***	***	
P92	Discrete Boundary Value 2,000	4.70E+00	434	129	7.13E+00	582	139	5.20E+00	518	372	6.543	425	232	3.74	227	132	1.421	197	147	
P93	Discrete Boundary Value 20,000	6.49E−01	0	0	7.61E−01	0	0	6.80E−01	0	0	1.022	0	0	0.57	0	0	0.562	0	0	
P94	Discrete Integral Equation 500	1.14E+01	181	34	1.25E+01	155	30	3.81E+00	61	16	4.359	59	13	3.93	61	17	3.733	59	18	
P95	Discrete Integral Equation 1,500	1.38E+02	236	46	1.28E+02	139	27	3.40E+01	59	14	43.34	63	16	43.8	74	15	34.87	61	19	
P96	Extended Powell Singular 1,000	***	***	***	***	***	***	***	***	***	***	***	***	1.55	413	132	2.033	948	640	
P97	Extended Powell Singular 2,000	***	***	***	***	***	***	***	***	***	***	***	***	12.3	889	324	8.542	1,259	845	
P98	Linear Full Rank 100	2.43E−01	128	18	1.25E+00	128	18	3.65E−02	63	13	0.085	63	13	0.06	63	13	0.089	65	16	
P99	Linear Full Rank 500	3.90E−01	114	15	3.87E−01	114	15	3.00E−01	84	18	0.409	84	18	0.23	72	15	0.157	65	14	
Note:

*** point of failure. Best results are in bold.

Table 4 Performance Comparison Based on CPU time (Tcpu), Number of function evaluation (NF), and number of iterations (Itr).

		RMIL	RMIL+	SRMIL	JYJLL	bib14	CG_DESCENT	
S/N	Functions/Dimension	Tcpu	NF	Itr	Tcpu	NF	Itr	Tcpu	NF	Itr	Tcpu	NF	Itr	Tcpu	NF	Itr	Tcpu	NF	Itr	
P100	Osborne 2 11	1.20E+00	8,569	2,719	6.29E−01	7,564	2,069	5.20E−01	8,743	6,791	***	***	***	0.05	674	375	0.17	1,821	1,517	
P101	Penalty1 200	5.19E−01	833	174	4.05E−01	1,643	314	7.62E−01	6,823	4,065	***	***	***	0.03	294	86	0.047	248	88	
P102	Penalty1 1,000	1.53E+01	2,423	364	2.43E+01	2,918	473	1.02E+01	1,674	892	***	***	***	1.55	297	78	1.334	343	130	
P103	Penalty2 100	2.88E−01	1,556	344	4.81E−01	1,456	332	9.02E−02	896	329	0.137	880	337	0.07	476	162	0.087	495	249	
P104	Penalty2 110	1.81E−01	1,260	243	1.56E−01	1,007	198	5.28E−02	460	173	0.284	576	202	0.06	556	128	0.089	890	249	
P105	Extended Rosenbrock 500	7.76E−01	741	179	1.11E+00	785	192	3.03E+00	2,588	1,871	***	***	***	0.3	289	61	0.354	354	196	
P106	Extended Rosenbrock 1,000	2.85E+00	611	126	6.43E+00	1,108	199	1.89E+01	3,731	2,637	***	***	***	1.61	363	84	1.186	384	205	
P107	Broyden Tridiagonal 500	3.24E-01	348	100	5.53E-01	475	125	1.25E-01	127	67	0.141	115	52	0.2	154	80	0.091	98	45	
P108	Broyden Tridiagonal 50	2.74E-02	345	63	1.23E-01	328	67	1.09E-02	100	45	0.179	95	40	0.01	114	53	0.006	98	44	
P109	HIMMELH 70,000	5.32E+00	1,158	126	1.08E+01	2,196	250	4.84E+00	1,032	114	0.952	155	33	***	***	***	5.597	1,221	141	
P110	HIMMELH 240,000	1.39E+01	859	107	9.92E+00	606	77	1.50E+01	918	105	2.291	105	23	***	***	***	8.097	512	79	
P111	Brown Badly Scaled 2	***	***	***	***	***	***	***	***	***	***	***	***	***	***	***	***	***	***	
P112	Brown and Dennis 4	9.00E−02	2,463	320	2.73E+00	5,922	752	6.20E−02	2,455	367	0.066	1,125	266	0.08	3,248	410	0.093	3,128	522	
P113	Biggs EXP6 6	***	***	***	***	***	***	***	***	***	***	***	***	0.06	1,513	582	***	***	***	
P114	Osborne1 5	***	***	***	***	***	***	***	***	***	***	***	***	***	***	***	***	***	***	
P115	Extended Beale 5,000	8.30E−01	713	199	8.00E−01	743	188	4.81E−01	427	226	0.883	647	369	***	***	***	0.232	180	106	
P116	Extended Beale 10,000	***	***	***	***	***	***	4.19E+00	1,710	1,240	***	***	***	55.7	543	153	0.493	218	128	
P117	HIMMELBC 500,000	1.19E+00	192	40	1.62E+00	306	57	6.74E−01	107	28	0.892	110	29	***	***	***	0.657	107	27	
P118	HIMMELBC 1,000,000	3.47E+00	313	57	3.18E+00	297	51	1.47E+00	110	29	1.713	108	28	***	***	***	1.26	107	28	
P119	ARWHEAD 100	***	***	***	7.79E−03	229	38	6.99E−03	83	25	0.003	74	25	0.01	141	32	0.011	94	49	
P120	ARWHEAD 1,000	***	***	***	***	***	***	6.48E−03	80	20	0.006	99	27	0.12	104	28	0.004	144	57	
P121	ENGVAL1 500,000	4.15E+01	5,572	560	2.95E+01	4,583	464	1.69E+01	2,349	244	10.51	1,048	113	***	***	***	9.476	1,384	166	
P122	ENGVAL1 1,000,000	1.34E+02	8,832	873	3.46E+01	2,704	280	4.60E+01	3,109	323	29.81	1,647	182	***	***	***	24.87	1,757	198	
P123	DENSCHNA 500,000	1.64E+01	239	43	2.04E+01	305	57	8.74E+00	126	51	9.435	112	37	***	***	***	7.983	116	51	
P124	DENSCHNA 1,000,000	2.95E+01	225	42	2.75E+01	208	43	1.82E+01	129	50	17.94	109	35	***	***	***	13.93	102	36	
P125	DENSCHNB 500,000	1.12E+00	192	38	1.22E+00	265	47	7.11E-01	113	29	0.77	106	27	***	***	***	0.637	109	32	
P126	DENSCHNB 1,000,000	2.16E+00	228	41	2.27E+00	222	39	1.44E+00	120	28	1.552	111	23	***	***	***	1.124	104	28	
P127	DENSCHNC 10	***	***	***	***	***	***	6.84E−03	128	43	0.005	128	38	0.01	175	36	***	***	***	
P128	DENSCHNC 500	***	***	***	***	***	***	***	***	***	0.025	170	62	***	***	***	***	***	***	
P129	DENSCHNF 500,000	***	***	***	***	***	***	6.29E+01	6,945	2,076	12.01	1,161	343	***	***	***	2.992	336	224	
P130	DENSCHNF 1,000,000	2.94E+02	17,218	4,175	***	***	***	7.25E+00	444	125	28.44	1,366	387	***	***	***	13.02	703	468	
P131	ENGVAL8 500,000	3.79E+01	5,159	516	8.53E+01	12,358	1,202	2.19E+01	2,824	281	24.38	2,557	263	***	***	***	39.6	5,514	572	
P132	ENGVAL8 1,000,000	5.81E+01	4,183	409	3.48E+02	25,510	2,415	7.38E+01	4,744	465	88.63	4,731	466	***	***	***	47.45	3,347	352	
Note:

*** point of failure. Best results are in bold.

To evaluate and compare the performances of all the methods, we use the Dolan & Moré (2002) performance metrics tool. For every formula, the tool graphs the fraction ρs(τ) of every problem solved based on the number of iterations (NOI) as shown in Fig. 1, number of functions evaluation (NOF) presented in Fig. 2 and CPU time given in Fig. 3. Note that the right side denotes the test functions successfully completed all the algorithms while the left-hand side of every graph represents the test problem percentage which defines the algorithm with better performance. The upper curve is the method with the best performance.

Figure 1 Performance metrics for NOI.

Figure 2 Performance metrics for NOF.

Figure 3 Performance metrics for CPU.

Observing all these graphs, we can see that the SRMIL formula is better than the RMIL, RMIL+, spectral JYJLL, NTTCG, and CG DESCENT methods because it converges faster and can complete more of the test functions.

Image restoration problem

This section investigates the performance of all the methods on image restoration models by recovering the original image from noisy or degraded versions. This is a specific form of unconstrained optimization problem, where the goal is to minimize the difference between the restored and original images. These problems are currently gaining a lot of attention in the research world due to their important applications in health and security (Malik et al., 2023; Yuan, Lu & Wang, 2020; Yu, Huang & Zhou, 2010; Salihu et al., 2023a). For this study, we consider restoring two images: CANAL (512×512) and GARDEN (512×512) that were corrupted by salt-and-pepper impulse noise. The quality of the restored image would be measured using peak signal-to-noise ratio (PSNR), relative error (RelErr), and CPU Time. The PSNR computes the ratio between the power of corrupting noises affecting fidelity of the representation and the maximum possible power of a signal. This metric is usually employed for measuring the qualities between the original and resultant images. A method with a higher PSNR value has better quality output images (Nadipally, 2019). The PSNR is computed as follows:

(27) PNSR=10⋅log10(MAXI2MSE)=20⋅log10(MAXIMSE)=20⋅log10(MAXI)−10⋅log10(MSE)

where MSE denotes the mean square error used for assessing the average pixel differences for complete images and MAXI represents the image’s possible maximum pixel value. A high MSE value designates a greater difference between processed and original images. Yet, the edges have to be carefully considered. The MSE is computed as follows:

(28) MSE=1N∑∑(Ei,j−oi,j)2,

where E denotes the edge image, o and N defines the original image and image sizes, respectively.

Let x define the real image with M×N pixel, the image restoration problem is modeled into an optimization problem of the form:

minH(u),

and

H(u)=∑(i,j)∈G{∑(m,n)∈Ti,j/Gϕα(ui,j−ξm,n)+12∑(m,n)∈Ti,j∩Gϕα(ui,j−um,n)},

where G denotes the index set of noise candidates x define as:

(29) G={(i,j)∈Q|ξ¯ij≠ξij,ξij=sminorsmax}.

Here, i,j∈Q={1,2,⋅,M}×{1,2,⋅,N} whose neighborhood is define as Tij={(i,j−1),(i,j+1),(i−1,j),(i+1,j)}, smax and smin represent the maximum and minimum of a noisy pixel. The observed noisy image corrupted by salt-and-pepper impulse noise is given as ξ and ξ¯ is the adaptive median filter of ξ.

Also, ϕα in H(u) denotes an edge-preserving potential function define as

(30) ϕα(t)=t2+α

where α is a constant whose value is chosen as 1.

The performance results from the computational experiments are presented below.

Table 5 compares the performance of four methods (SRMIL, RMIL, RMIL+, and JYJLL) on image restoration models across different noise levels (30%, 50%, and 80%). Two images (“CANAL” and “GARDEN”) are considered, with CPU time (CPUT), relative error (RelErr), and peak signal-to-noise ratio (PSNR) used as evaluation metrics. SRMIL stands out as the fastest method, particularly at lower noise levels, where it provides competitive restoration results with minimal computational cost. While it shows slightly higher relative errors and lower PSNR at higher noise levels compared to RMIL+ and RMIL, it remains a highly efficient choice for scenarios where speed is critical. RMIL+ delivers a competitive performance at higher noise levels in terms of PSNR and relative error, but comes with a higher computational cost. RMIL offers a middle ground with moderate performance, while JYJLL is the least efficient in both quality and speed. Overall, SRMIL is the most efficient choice for speed and quality trade-offs, especially at lower noise levels, making it a strong option for time-sensitive applications.

Table 5 Image restoration outputs for SRMIL, RMIL, RMIL+, and JYJLL based on CPUT, RelErr and PSNR metrics.

METHOD	SRMIL	RMIL	RMIL+	JYJLL	
IMAGE	NOISE	CPUT	RelErr	PSNR	CPUT	RelErr	PSNR	CPUT	RelErr	PSNR	CPUT	RelErr	PSNR	
CANAL	30%	48.4348	0.8080	31.4865	48.4444	0.7969	31.2218	48.5786	0.8270	31.2973	48.8289	0.7970	31.3173	
	50%	131.7956	1.3664	28.0092	104.9128	1.3886	27.7759	107.4118	1.2796	27.8678	137.1973	1.2892	27.7886	
	80%	168.4606	2.1083	24.8027	200.8903	2.6569	23.1556	195.5230	2.6070	23.1988	196.3052	2.6798	23.1988	
GARDEN	30%	63.5339	1.1222	28.5409	63.5676	1.1348	28.3746	62.6936	1.1634	28.3828	66.3669	1.1752	28.4827	
	50%	128.4747	1.5731	25.5737	105.3938	1.7243	25.3806	227.0897	1.7072	25.2218	218.8192	1.7127	25.1822	
	80%	163.2581	2.2388	22.9950	195.6410	2.6818	21.7677	195.5568	2.7264	21.7584	181.9556	2.1472	21.8475	

The performance of SRMIL, RMIL, RMIL+, JYJLL was evaluated based on CPU Time, RelErr, and PSNR, as summarized in Table 5 with the graphical representation of the best performers presented in Figs. 4, 5, 6 respectively. It can be seen that all the methods are successful on all the perception-based quality metrics discussed earlier. However, the proposed method is acknowledged as the best performer because it produced the least CPU time and higher PSNR values for all the noise degrees. This is supported by the fact that a method with higher PSNR values produces better quality of the output images.

Figure 4 CANAL & GARDEN images corrupted by 30% salt-and-pepper noise: (A, F), the restored images using SRMIL: (B, G), RMIL: (C, H), and RMIL+: (D, I), and JYJLL: (E, J).

Figure 5 CANAL & GARDEN images corrupted by 50% salt-and-pepper noise: (A, F), the restored images using SRMIL: (B, G), RMIL: (C, H), and RMIL+: (D, I), and JYJLL: (E, J).

Figure 6 CANAL & GARDEN images corrupted by 80% salt-and-pepper noise: (A, F), the restored images using SRMIL: (B, G), RMIL: (C, H), and RMIL+: (D, I), and JYJLL: (E, J).

Therefore, this demonstrates the importance of the proposed algorithm in solving some applications as well as highlighting some challenges of developing new iterative algorithm for real-life problems.

Application in robotic arm motion control

Robotic motion control are processes and systems employed to control the movement of robots. This field encompasses different technologies and techniques to ensure efficient and accurate movement of robotic systems. Some real-world applications of these robots include industrial automation where the robots perform tasks such as packaging, painting, welding, and assembly. Also, service robots assist in hospitality or household chores while medical robots include rehabilitation devices and surgical robots. The major problem of robotic motion control often arises as a result of various factors including dynamic uncertainties, actuator limitations, sensor noise, and environmental disturbances. For instance, the robot may fail to follow the desired path accurately due to non-linearities, delays, disturbances, and inaccurate modeling. This problem can be overcome by adaptive control to adjust the parameters in real time or by using advanced control techniques such as the currently used gradient-based methods.

In this study, the application of the proposed gradient-based SRMIL algorithm will be demonstrated on a simulated robotic model. Our approach leverages the conjugate gradient (CG) method within an inverse kinematics framework to optimize joint angles to achieve desired end-effector positions. Specifically, we solve non-linear systems formulated from the kinematic equations of the robotic arm, where the proposed CG method is used to iteratively refine joint parameters to minimize positioning error.

The robotic model, originally defined with two degrees of freedom (2DOF) as detailed in Sulaiman et al. (2022) and Awwal et al. (2021), has currently been extended to include three degrees of freedom (3DOF) (Yahaya et al., 2022; Yunus et al., 2023). Noted that the scope of this study for robotic motion control is limited to a simulated 3DOF robotic system. This system is designed to evaluate the algorithm’s capability to handle trajectory optimization and precision control challenges. The problem description begins with an illustration of the planar three-joint kinematic model equation and the formulation of the discrete kinematic model equation for three degrees of freedom as presented in Yahaya et al. (2022), Zhang et al. (2019), Yunus et al. (2023).

(31) G(ηk)=ρk,

where ρk∈R2 and ηk∈R2 denote the end vector effector position and the joint angle vector respectively. with ηk∈R2 denoting the joint angle vector, ρk∈R2 defining the end vector effector position, and the function G(⋅) representing the kinematic mapping whose equation is formulated as follows:

(32) G(ηk)=[τ1cos⁡(η1)+τ2cos⁡(η1+η2)+τ3cos⁡(η1+η2+η3)τ1sin⁡(η1)+τ2sin⁡(η1+η2)+τ3sin⁡(η1+η2+η3)],

with τ1, τ2, and τ3 represent the first, second and their rod lengths respectively. The function G(⋅) is responsible for mapping the active joint displacements to any part of the robot ηk∈R2 or position and orientation of a robot’s end-effector. In the case of robotic motion control, G(η) represents the end-effector position vector. To achieve the problem formulation at a specific moment tk, we define the preferred path vector as μtk∈R2. Based on the preferred path vector, we can now formulate the following least-squares problem which is addressed at every interval tk∈[0,tf]:

(33) minη∈R212||G(η)−μtk||2,

where μtk denotes the end effector controlled track and as described in Yahaya et al. (2022), the Lissajous curves route required at time tk are generated using:

(34) μtk1=[32+15sin⁡(πtk5)32+15sin⁡(2πtk5+π3)].

(35) μtk2=[32+15sin⁡(4tk)32+15sin⁡(3tk)].

(36) μtk3=[32+15sin⁡(2tk)32+15sin⁡(tk)].

Observe that the structure of Eq. (33) is similar to that of 1 defined earlier which enables utilizing the proposed SRMIL algorithm to solve this problem. For the experiment simulation, the parameters used in the implementation are τ1=1, τ2=1, τ3=1, and tf=10 s. Each link is connected by a rotational joint, enabling motion within a two-dimensional workspace. The starting point ηt0=[η1,η2,η3]=[0,π3,π2]T with the duration of task [0,10] divided into 200 equal segments, providing a structured timeline for the optimization process and allowing precise evaluation of the method’s performance over time. The joint angles η1, η2, and η3 determine the configuration of the arm, and the position of the end effector is denoted by ρk∈R2. This planar configuration ensures that the arm can achieve various positions within its workspace, constrained by the total link lengths and joint angles. The following algorithm describe the procedure used in solving the problem.

Algorithm 2 Implementation of SRMIL method on 3DOF robotic arm model.

 Step 1: Inputs: Initialize parameters t0, ηt0, tmax, g, and Kmax	
 Step 2: For k=1 to Kmax do	
  tk=k∗g;	
 Step 3: Evaluate Lissajous curves μtk1n,n=1,2,3 using Eqs. (34), (35), (36);	
 Step 4: Compute ηtk using SRMIL (ηt0,μtk,i(n));	
 Step 5:  Set ηnew=[ηt0;ηtk];	
 Step 6: Output: ηnew.	

The experimental results of the SRMIL for motion control are demonstrated in Figs. 7–10.

Figure 7 End effector trajectory and desired path (A) and Synthesized robot trajectories of Lissajous curve (Eq. (34)).

Figure 8 End effector trajectory and desired path (A) and Synthesized robot trajectories of Lissajous curve (Eq. (35)).

Figure 9 End effector trajectory and desired path (A) and Synthesized robot trajectories of Lissajous curve (Eq. (35)).

Figure 10 Tracking the residual error of Lissajous curve along x axes (A), along y axes (B).

The detailed explanation of the results is as follows. Figures 7A, 8A, and 9A demonstrate the robot’s end effector model precisely following the desired path of Lissajous curve Eqs. (34)–(36), respectively. While Figs. 7B, 8B, and 9B illustrate the synthesis of the robot’s trajectories of the Lissajous curve Eqs. (34), (35) and (36), respectively. Figures 10A and 10B represent the error rates of residuals, indicating that SRMIL recorded the least errors despite the close competition the method faced from the classical CG-DESCENT algorithm. The higher residual errors observed in existing methods might be due to limitations in their ability to handle nonlinearities in robotic motion control tasks. On the other hand, the low residual error rates recorded by the proposed SRMIL algorithm can be attributed to its ability to efficiently handle the complexities and dynamics requirements of the considered problems, and thus, demonstrate its efficiency and robustness in practical application.

The study further compared the performance of the algorithms including SRMIL, JYJLL, RMIL+, and RMIL on Lissajous curve problems (see; Table 6), using computational time (CPUT) and iteration count (Iter) as evaluation metrics. For clarity, we bolded the best results to easily differentiate the performance of the algorithms.

Table 6 Performance comparison of computational time and number of iteration for SRMIL, RMIL, RMIL+, and JYJLL base on different Lissajous curves.

METHOD	SRMIL	JYJLL	RMIL+	RMIL	
LISSAJOUS CURVES	EQUATION	CPUT	Iter	CPUT	Iter	CPUT	Iter	CPUT	Iter	
μtk1	(34)	0.3839	64	0.3972	81	0.3672	62	0.3893	65	
μtk2	(35)	0.3592	27	0.4000	30	0.3688	29	0.3597	27	
μtk3	(36)	0.3175	38	0.4246	43	0.3173	38	0.3149	37	
Note:

Best results are in bold.

As observed in Table 6, the proposed SRMIL demonstrates strong performance, achieving competitive or superior results in both metrics across all cases. RMIL+ closely rivals SRMIL, especially for μtk1, where it achieves the fastest computational time and fewest iterations. RMIL shows consistent, reliable performance, slightly trailing SRMIL and RMIL+ in most cases. On the other hand, JYJLL is the least efficient, with higher iteration counts and computational times across all problems. Overall, SRMIL and RMIL+ emerge as the most effective algorithms, excelling in both iteration count and efficiency, while RMIL remains a strong alternative. JYJLL’s relatively poor performance indicates it may require further refinement for problems of this type. These results further demonstrate the potential of SRMIL and RMIL+ for solving optimization problems involving Lissajous curves, with SRMIL offering the best balance of computational efficiency and convergence behavior.

Conclusions

In this article, we have presented a descent modification of the RMIL+ CG algorithm such that the coefficient βk does not become superfluous. The proposed method was further extended to construct a new search direction that guarantees a sufficient decrease in the objective function. The global convergence was discussed using the Lipschitz continuity assumption. Numerical results on a range of test problems were reported to evaluate the algorithm’s performance. Specifically, the new method demonstrated consistent effectiveness by outperforming other algorithms, including CG-DESCENT, in terms of iteration counts and function evaluations. Robustness was assessed by testing the algorithm on problems with varying dimensions and initial points. Additionally, the practical applicability of the proposed algorithm was validated through detailed comparisons in image restoration and 3DOF robotic motion control simulation involving a generic 3DOF system designed to capture typical challenges in trajectory optimization. Future work will involve additional tests of the proposed algorithm, including sensitivity analyses and real-world scenarios, such as experiments with actual robotic systems, to further substantiate its robustness.

Supplemental Information

Supplemental Information 1 Experimental code.

The authors express their gratitude to the reviewers for their valuable comments and suggestions.

Additional Information and Declarations

Competing Interests

The authors declare that they have no competing interests.

Author Contributions

Sulaiman Mohammed Ibrahim conceived and designed the experiments, performed the computation work, prepared figures and/or tables, and approved the final draft.

Aliyu M. Awwal conceived and designed the experiments, performed the computation work, authored or reviewed drafts of the article, and approved the final draft.

Maulana Malik conceived and designed the experiments, analyzed the data, performed the computation work, prepared figures and/or tables, and approved the final draft.

Ruzelan Khalid performed the experiments, prepared figures and/or tables, and approved the final draft.

Aida Mauziah Benjamin performed the experiments, authored or reviewed drafts of the article, and approved the final draft.

Mohd Kamal Mohd Nawawi conceived and designed the experiments, authored or reviewed drafts of the article, and approved the final draft.

Elissa Nadia Madi performed the experiments, analyzed the data, authored or reviewed drafts of the article, and approved the final draft.

Data Availability

The following information was supplied regarding data availability:

The data are available in the Tables and code is available in the Supplemental Files.

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
