# Peer review of "An efficient gradient-based algorithm with descent direction for unconstrained optimization with applications to image restoration and robotic motion control"

_PeerJ Computer Science, doi:10.7717/peerj-cs.2783_

## Round 0.1 · original submission · Minor Revisions

Dear authors,

You are advised to critically respond to all comments point by point when preparing an updated version of the manuscript and while preparing for the rebuttal letter. Please address all comments/suggestions provided by reviewers, considering that these should be added to the new version of the manuscript.

Kind regards,
PCoelho

Reviewer 1 ·

Basic reporting

The basic content of the article is adequate, it also presents an important compendium of publications that form the background, however, the authors do not highlight the contribution of their work and in the introduction they do not explain the development of their work.

Experimental design

The authors present two cases, that image reconstruction and that controlling a 3 DOF robotic arm. In the latter case, the authors do not mention the procedure for implementing the algorithm in controlling the robot. Likewise, it is not clear. What control technique do they use?

Validity of the findings

It is necessary to explain in more detail the results of figures 4, 5 and 6 of the image reconstruction, since they look the same. It is also necessary to explain the procedure for implementing the algorithm to robotic control and explain in detail the residual errors shown in Figure 10.

Additional comments

In introduction section, authors show an appropriate reference of important background information, however, the authors do not describe the development of the work, they do not clearly mention the problem to be solved to justify their work and it is necessary to highlight the contributions they have in the work.
In results shown in figures 4, 5 and 6, being a visual perception, it is difficult to identify the best results of the proposed algorithm. The authors are recommended to explain the differences in each of the restored images. On the other hand, in Table 1 the authors summarize the evaluation of the different methods evaluated, it is recommended that the authors highlight the best results of each of the methods.
In position or movement control tests, on a 3D robot arm, they show 3 cases, I suppose it is with the srmill algorithm, and in figure 10 they show the errors in the x and y axes, however, it is not clear that it has smaller error than the other algorithms, it is suggested that the authors find a better way to display these results.
In conclusions section, authors mention that they have demonstrated the effectiveness and robustness of their proposed algorithm, however, basic tests are not enough to demonstrate effectiveness and much less to demonstrate robustness without clarifying robustness with respect to that?
It is necessary that the authors give more details to what they refer to with practical applicability, since the results only apply to two cases, and that of the robot arm is in simulation and very general without giving details of the type of robot.
Application of the proposed algorithm in unrestricted optimization problems is not clear, as the authors announce in the title.

Reviewer 2 ·

Basic reporting

- The paper is generally written in professional English, and the technical language is appropriate for a specialized audience.

- Sufficient field background and context are provided, making it easy to follow.

Experimental design

- The manuscript presents a clear research question, focusing on improving the conjugate gradient (CG) algorithm to address challenges in optimization for image restoration, robotic motion control, and other unconstrained optimization problems. This is a relevant and meaningful question within the journal’s scope.

- However, the authors should explain why the SRMIL method is needed for robotic arm motion control, especially when classical CG methods can solve the problem well.

Validity of the findings

- The authors mentioned, "However, the proposed method is acknowledged as the best performer because it produced the least CPU time and higher PSNR values for all the noise degrees." Please expand on this, as CPU time and PSNR are not always better for SRMIL.

- For Figure 10, it would be helpful to clarify why residual errors are higher for other methods. Additionally, providing computational time comparisons for this case study across different methods would add meaningful insights.

- Furthermore, the authors mentioned in the conclusion that "the proposed algorithm is significantly more effective than other methods, including the famous CG-DESCENT algorithm." Please expand on this.

Additional comments

- Please explain what 0-200 means in Figure 10.

- Please explain the geometry/link lengths of the robotic arm, as this information is necessary.

Reviewer 3 ·

Basic reporting

A new gradient-based algorithm is developed to improve the performance of optimization models related to image restoration. The objective function is decreasing. The algorithm's convergence is based on Lipscthiz continuity assumptions. Concrete examples show better restoration results than other methods.

We noticed that many references are not cited. They should be cited or deleted.

Experimental design

Same as above

Validity of the findings

Same as above

Additional comments

Same as above

---

## Round 0.2 · Minor Revisions

Dear authors,

Thanks a lot for your efforts to improve the manuscript.
Nevertheless, some concerns are still remaining that need to be addressed.
Like before, you are advised to critically respond to the remaining comments point by point when preparing a new version of the manuscript and while preparing for the rebuttal letter.

Kind regards,
PCoelho

Reviewer 1 ·

Basic reporting

Authors have attended to a large part of the observations, they have highlighted the contributions of their work, however, the way of showing the results could be improved.

Experimental design

In the observations sent to the authors, they are asked about the control techniques used in their work. However, his answer is not clear.

Validity of the findings

Authors have addressed the observations made in this section

Additional comments

Authors have addressed the vast majority of the observations sent in this section, however, in comment 3 it was suggested to highlight the best results in the tables (table 1) and in comment 4 it was suggested to improve the presentation of results in the Figure 10, in both cases, it is not evident that the authors have adequately addressed these suggestions.

---

## Round 0.3 · accepted · Accept

Dear authors, we are pleased to verify that you meet the reviewer's valuable feedback to improve your research.

Thank you for considering PeerJ Computer Science and submitting your work.

Kind regards
PCoelho

Reviewer 1 ·

Basic reporting

Authors have addressed the observations, they have highlighted the better results shown in Tables 1-4, however, the detailed explanation of these better results is still lacking.

Experimental design

In the comments made to the authors, they were questioned about the control techniques used, they have already included a writing indicating that they used a controller based on the conjugate gradient approximation.

Validity of the findings

In this section the observations had already been addressed in the previous review.

Additional comments

Authors have addressed the observations made in the previous review, however, the presentation of the best results shown in Tables 1 to 4 would be better highlighted with a more detailed explanation in the text.